# P-tau217 as a Biomarker in Alzheimer’s Disease: Applications in Latin American Populations

**DOI:** 10.3390/ijms26146633

**Published:** 2025-07-10

**Authors:** Christian Alexis Varela-Vidales, Alejandra Martínez-Hernández, Elizabeth Hernández-Castellanos, Daniela L. C. Delgado-Lara

**Affiliations:** 1Departamento Académico de Formación Universitaria, Ciencias de la Salud, Universidad Autónoma de Guadalajara, Zapopan 45129, Jalisco, Mexico; christian.varela@edu.uag.mx (C.A.V.-V.); martinezh.alejandra@edu.uag.mx (A.M.-H.); 2Unidad-321 Zacatecas, Universidad Pedagógica Nacional, Guadalupe 98615, Zacatecas, Mexico; 3School of Dietetics and Human Nutrition, McGill University, Montreal, Sainte-Anne-de-Bellevue, Québec, QC H9X 3V9, Canada; elizabeth.hernandezcastellanos@mail.mcgill.ca

**Keywords:** Alzheimer’s disease, p-tau217, biomarker, Latin America

## Abstract

Alzheimer’s disease (AD) is one of the primary dementia causes worldwide. For this reason, there is a need for plasma-based diagnostic biomarkers to facilitate the timely diagnosis of AD. This work synthesizes the current evidence concerning the tau protein p-tau phosphorylated at threonine 217 (p-tau217) as an emerging biomarker, emphasizing its utility in preclinical phases and its potential application in Latin American populations. The findings indicate that p-tau217 has superior sensitivity and specificity compared to classical biomarkers such as p-tau181 and Aβ42. Likewise, its plasma concentration regulates neuropathological progression, as studies by Braak have shown, enabling it to identify alterations from the early stages. In Latin America, studies in Peru, Colombia, and Brazil have shown promising results, albeit with methodological limitations. Some of them have small sample sizes or lack neuroimaging confirmation. Additionally, clinical factors common in the region, such as hypertension, diabetes, or chronic kidney disease, may alter the clinical interpretation. In short, p-tau217 represents a potential non-invasive diagnostic resource. More diverse cohorts are needed to confirm its validity in daily clinical practice.

## 1. Introduction

Alzheimer’s disease (AD) is a progressive neurodegenerative disorder considered the leading cause of dementia worldwide. Currently, it is estimated that more than 55 million people have dementia; this is 5–7% of the population older than 60 years [1]. It is estimated that around 10% of adults older than 60 years will develop AD [2]. Projections indicate that this prevalence will triple by 2050 due to the increasing life expectancy and global aging population trends [2]. In Latin America, dementia is a growing concern for public health, with a prevalence higher than in the US and Europe (7.1% to 11.55%) [3]. In addition, it is estimated that 58% of the people with dementia live in low- and middle-income countries, and it is predicted that this proportion will continue to increase [3].

From a pathological standpoint, AD is characterized by the accumulation of beta-amyloid (Aβ) plaques and neurofibrillary tangles composed of hyperphosphorylated tau protein (p-tau), whose accumulation is associated with progressive loss of synaptic function and eventual neurodegeneration [4]. This tau hyperphosphorylation occurs at multiple sites, altering its normal function of stabilizing microtubules and promoting its aggregation [5]. Therefore, the phosphorylation at threonine residue site 217 (p-T217) becomes relevant in this pathology [4,6].

Traditionally, a definitive AD diagnosis has been made through histological post-mortem analysis. However, it has been documented that pathological alterations can begin up to twenty years before the onset of clinical symptoms [6]. The early detection of AD is fundamental for a better prognosis and timely treatment to preserve the patient’s quality of life [7]. Preclinical detection provides a unique opportunity for intervention and potentially slowing the progression of the disease before symptoms become clinically evident [8]. Therefore, research on plasma biomarker quantification stands as a priority. It represents a minimally invasive and feasible alternative to cerebrospinal fluid (CSF) analysis, enabling its use in population-level screening strategies [9].

Among the emerging plasma biomarkers, p-tau phosphorylated at threonine 217 (p-tau217) has gained recognition for its high specificity and sensitivity in detecting neurodegenerative changes associated with AD [10,11]. Recent studies have shown that the p-tau217 levels increase during the preclinical stages of the disease, allowing the identification of individuals at a high risk of developing AD [10,11,12,13]. Thus, it is essential to evaluate the clinical potential of p-tau217, both as a screening tool and for the longitudinal monitoring of disease progression [11,14]. Therefore, this review aimed to synthesize the existing evidence concerning p-tau217 as a biomarker for the detection and monitoring of AD, with a focus on the Latin American population.

## 2. Methodology

This narrative review was conducted according to the SANRA (Scale for the Assessment of Narrative Review Articles) recommendations. The literature search was conducted in PubMed from January 2020 to May 2025, utilizing a combination of Medical Subject Headings (MeSH), DeCS descriptors, and free-text terms. The keywords included “p-tau217”, “Alzheimer’s disease”, “biomarkers”, “Braak stages”, “early diagnosis”, “CSF”, “plasma”, “p-tau217 Latin America”, “comorbidities”, “diabetes”, “renal disease”, “depression”, and “Alzheimer’s disease Latin American populations”. Boolean operators (AND/OR) were used to refine the results by thematic axis.

The search strategy evolved iteratively as the scope of the review narrowed to focus on the diagnostic utility of p-tau217 in Latin American populations with comorbidities. Titles, abstracts, and full texts were screened manually by all the authors to identify relevant studies. Both original research articles and reviews were included to ensure a comprehensive understanding of the topic. Only publications in English and human studies were considered. Animal model studies or those lacking clinical or biomarker data were excluded.

## 3. The Role of Tau and P-tau217 in Alzheimer’s Pathophysiology

### 3.1. Amyloidogenic and Non-Amyloidogenic Pathways of APP Processing

AD is characterized by the accumulation of neuritic and beta-amyloid (Aβ) plaques, along with neurofibrillary tangles, in the brain. These pathological changes are followed by neuronal loss, particularly of cholinergic neurons in the basal forebrain and neocortex [15]. According to the amyloid hypothesis, it has been suggested that the Aβ plaques originate from the amyloid precursor protein (APP) through sequential cleavage by the β- and γ-secretase enzymes, triggering a pathological cascade associated with AD [16].

The choice between the non-amyloidogenic pathway (physiological) and the amyloidogenic pathway (pathological) depends on enzymatic accessibility and secretase modulation. Thus, this duality positions APP as a critical upstream regulator [17,18,19,20].

In the non-amyloidogenic pathway, APP is cleaved by α-secretase, a metalloprotease belonging to a disintegrin and metalloproteinase 10 (ADAM10) family, within the Aβ domain, avoiding the formation of the β-amyloid peptide (Aβ) [17,20]. This pathway generates two fragments: soluble APP alpha (sAPPα), which is secreted into the extracellular space, playing neurotrophic and synaptic roles, and the C-terminal fragment of 83 amino acids, which is further cleaved by ɣ-secretase, releasing a short non-amyloidogenic peptide (p3) and the APP intracellular domain [17,18,20].

Conversely, in the amyloidogenic pathway, APP is first cleaved by β-secretase, which generates two fragments: soluble APP beta (sAPPβ) and the 99-amino-acid C-terminal fragment of APP (C99) [18]. ɣ-Secretase subsequently cleaves C99, thus liberating Aβ, in its 40- and 42-amino-acid forms (Aβ40 and Aβ42), along with the APP intracellular domain [18,19,20]. While Aβ40 is the most abundant isoform, Aβ42 is more hydrophobic and prone to aggregation, which promotes the development of amyloid plaques [21].

The activation of these pathways depends on the compartmentalization of APP, secretase maturation, the content of membrane cholesterol, and the transcriptional regulation of related genes, among other factors [17,20]. The development of biomarkers in CSF is gaining momentum for the early detection of AD due to its hallmark characteristics, including disruptions in APP processing, activation of the amyloidogenic pathway, and Aβ42 accumulation.

### 3.2. Tau as a Mediator of Neurodegeneration

The decrease in Aβ42 reflects its deposition in the brain parenchyma, while an increase in the total tau (t-tau) and phosphorylated tau at threonine 181 (p-tau181) levels in CSF indicates active neurodegeneration [22,23]. These three biomarkers are the classical biomarker profile in AD [24]. This profile is generally employed for research and clinical purposes. However, the sensitivity and specificity of this basic profile have shown several limitations, particularly in preclinical stages, which has prompted the search for new, more precise biomarkers, such as p-tau217 [25].

The tau protein is a microtubule-associated phosphoprotein (MAP) that is expressed mainly in the neurons of the central nervous system [26]. In physiological conditions, tau stabilizes the neuronal cytoskeleton and participates in the axonal transport of organelles, vesicles, and proteins that are essential for the maintenance of neuronal function [26].

There are six isoforms of tau derived from the alternative splicing of the MAPT gene (17q21 locus on chromosome 17) [27]. These isoforms differ in the number of microtubule-binding repeat domains, either three or four, a feature that influences their affinity for microtubules and their aggregation propensity [28,29].

From a structural perspective, the tau protein has multiple residues susceptible to phosphorylation, mainly in the regions adjacent to the microtubule-binding domain [28]. In AD, tau is hyperphosphorylated at specific residues, thus decreasing its affinity for microtubules [28]. This promotes its dissociation, cytoplasmatic accumulation, and aggregation into neurofibrillary tangles [4,28]. Among the residues, threonine 217 (T217), located within the TPP motif (Thr-Pro-Pro), is an epitope sensitive to structural modifications, particularly phosphorylation, which gives as a result the isoform known as p-tau217 [30]. This process is illustrated in Figure 1.

### 3.3. P-tau217 as a Link Between Amyloid Pathology and Neuronal Degeneration

P-tau217 has a high specificity for AD detection, contrary to other isoforms such as p-tau181 or p-tau231, which distinguishes it from other tauopathies [13,25,30,31,32,33]. Studies in CSF and plasma have shown that p-tau217 is associated with cortical tau accumulation and amyloid positivity measured by tau-specific and amyloid-specific positron emission tomography (tau-PET and amyloid-PET), even in the preclinical stages [34]. In animal models with phosphomimetic mutations at T217, an increase in aggregate formation, synaptic damage, and accelerated cognitive decline have been observed [35].

Phosphorylation at T217 is primarily induced by the kinases glycogen synthase kinase three beta (GSK3β) and cyclin-dependent kinase 5 (CDK5), enzymes involved in the neurodegenerative cascade that activates in conditions of oxidative stress, Aβ accumulation, or synaptic dysfunction [36,37]. Additionally, the kinases p38 mitogen-activated protein kinase (p38 MAPK) and AMP-activated protein kinase (AMPK) might be involved in the cell phosphorylation in different pathological residues related to the progression of AD and indirectly contribute to the conformational changes of tau that favor microtubule instability, expose the hydrophobic domains, and facilitate the formation of neurotoxic oligomers [36,37,38,39].

The isoform of p-tau217 not only represents an early diagnostic biomarker but also plays a role in several pathological mechanisms of AD [34]. The extracellular oligomers of Tau (oTau) are involved in synaptic communication, which implies an immediate decline in the long-term potentiation and hippocampal memory formation, even with normal levels of Aβ [40]. Studies of oTau in the brain tissue of AD patients and animal tauopathy models show that these are internalized rapidly by the neurons, where they alter the postsynaptic signaling [40]. Furthermore, p-tau217 increases in response to oTau, which might contribute to the synaptic decline [41]. These findings demonstrate the intrinsic pathogenic role of oTau in AD, independent of amyloid pathology considerations. Thus, different therapeutic targets are revealed for the oligomeric species of tau [40].

At the mitochondrial level, the p-tau217 and phosphomimetics in threonine 231/serine 235 alter the dynamics of the mitochondria and impair the energetic efficiency of the neuron, which worsens the cell vulnerability in key cortical regions, where energetic demand is critical to sustain memory [42]. Histopathologic studies have identified the presence of p-tau217 in granulovacuolar degeneration bodies in subcellular compartments located in the neurons of the hippocampus CA1 [36]. These findings suggest that tau is implicated in the lysosomal dysfunction of AD [43].

An important aspect to consider in the pathophysiology of p-tau217 as a biomarker is its applicability in diverse populations. The current literature has been reported in North American and European populations. Therefore, it remains unknown whether differences exist in the phosphorylated tau expression across various genetic and environmental contexts, such as those characteristics of the Latin American population. Given the rising interest in personalized medicine strategies, understanding the peculiarities of each region in the pathophysiology of tau is key to validating and adapting p-tau217 use in different clinical contexts.

## 4. Diagnostic Performance of P-tau217 Compared to Other Biomarkers

### 4.1. Value of P-tau217 as a Biomarker for Disease Progression and Preclinical Diagnosis

P-tau217 is considered a useful screening tool to detect AD in the early stages, even before the onset of detectable cognitive symptoms as detected by tau-PET [10]. Several studies have evidenced that the plasma levels of tau increase early in healthy subjects with amyloid positivity (Aβ+), which reflects an initial phosphorylation activation of tau induced by Aβ [10,44,45].

In the BioFINDER-2 cohort, conducted at the Clinical Memory Research Unit, Lund University, Lund, Sweden Palmqvist et al. demonstrated the superior diagnostic performance of plasma p-tau217 compared to p-tau181 in distinguishing AD from other neurodegenerative disorders. The area under the curve (AUC) for p-tau217 was 0.96 (95% CI: 0.93–0.98), significantly higher than that of p-tau181 (AUC = 0.81; 95% CI: 0.74–0.87) [10]. Furthermore, an improvement was observed in the diagnostic performance in the preclinical stages in presenilin 1 (PSEN1) E280A mutation carriers from the Colombian autosomal dominant cohort [10]. This finding was replicated in the same Colombian cohort, where the p-tau217 levels increased during the presymptomatic phases and accurately distinguished Aβ-positive individuals from Aβ-negative controls [10].

In another longitudinal study derived from the same cohort, Palmqvist et al. reported that the plasma p-tau217 levels alone predicted AD progression to four years, with an AUC of 0.83 (95% CI: 0.78–0.89) [46]. This predictive power improved significantly when it was combined with memory and executive function assessments and apolipoprotein E (*APOE*) genotype data (AUC = 0.91; 95% CI: 0.87–0.94; *p* < 0.001) [46]. Even p-tau217 alone demonstrated greater predictive accuracy than a clinical judgment made by dementia specialists (AUC = 0.81 vs. AUC = 0.72; *p* = 0.03) [46].

From the pathophysiological standpoint, Mattsson-Carlgren et al. demonstrated that Aβ accumulation was associated with a significant rise in the p-tau217 levels in CSF (*p* < 0.0001), even before tau-PET detected the pathology. Many Aβ+ individuals presented increased levels of p-tau217 with a negative tau-PET [45]. This finding positions this biomarker as an early-stage indicator of the transition from the asymptomatic to the symptomatic stage of AD [45].

Ossenkoppele et al. proposed a sequential screening model in which plasma p-tau217 serves as a preselection tool to determine the need for tau-PET imaging in AD clinical trials. Using p-tau217 alone reduced the number of required tau-PET scans by 76%, while the sequential approach of applying tau-PET only in cases selected by the plasma levels achieved a 94% reduction [44]. This strategy significantly enhances the diagnostic efficiency in the preclinical stages of AD [44]. In Latin American countries, where access to tau-PET is limited due to high costs and technological constraints, this sequential model represents a key opportunity to promote earlier detection and more equitable recruitment in clinical trials. Ultimately, it may help to make AD diagnosis and research more accessible and sustainable in low-resource settings.

A multicenter European study demonstrated that plasma p-tau217, measured with the Fujirebio Lumipulse^®^ G1200 automated platform (Fujirebio Diagnostics, Inc., Malvern, PA, USA), detected AD with an AUC of between 0.93 and 0.96, both at the primary and secondary level of attention. The positive predictive value ranged from 82% to 95%, and the negative predictive value from 88% to 90%, supporting its utility outside of highly specialized settings [47].

Recently, in September 2024, Fujirebio Diagnostics, Inc. submitted to the FDA a request for approval of the Lumipulse^®^ G pTau217/β-Amyloid 1-42 plasma ratio immunoassay as the first blood test to support the diagnosis of AD [48]. In May 2025, the FDA granted 510(k) clearance, marking a milestone in the clinical implementation of non-invasive biomarkers for the early diagnosis of AD, particularly in settings where therapeutic intervention may be most effective [49].

### 4.2. Comparative Analysis with Other Alzheimer’s Disease Biomarkers

The comparison of p-tau217 and other biomarkers has positioned it as one of the most solid plasmatic biomarkers used to detect AD. This section analyzes its relative performance compared to CSF biomarkers, including Aβ42, t-tau, and p-tau181. The classical CSF biomarkers have been essential for the development of the AT(N) framework proposed by Jack et al., which classifies based on their accumulation (A), pathological tau (T) and neurodegeneration (N). Each of the indicators has clinical and analytic limitations that have driven the development of methods with greater sensitivity, specificity, and accessibility [1].

One of the CSF biomarkers that has demonstrated a high level of sensitivity and specificity for detecting clinical stages is Aβ42 [4,50]. However, its specificity is not absolute since reduced concentrations have also been observed in other conditions affecting amyloid metabolism, such as vascular disorders and mixed encephalopathies [51]. Furthermore, the variability among individuals and pre-analytic conditions, such as the type of collection tube, processing times, and storage conditions, also affect its stability and diagnostic interpretation [4,52].

The protein t-tau increases in the presence of neurodegeneration, but its diagnostic specificity is limited. Its levels increase in other pathologies, such as Creutzfeldt–Jakob disease, where the levels can be up to twenty times higher than those observed in AD without a concomitant increase in p-tau [1,23]. For this reason, t-tau is considered more of a nonspecific biomarker of neurodegeneration rather than a disease-specific hallmark of AD [1].

P-tau181 in CSF presents a greater specificity than t-tau for the diagnosis of AD. It does not present significant variations in other neurological conditions, and its correlation with neurofibrillary tangles has been backed up by neuroimaging and post-mortem studies [1,53]. Additionally, its levels are notably elevated from the preclinical stages in individuals with diagnosed cerebral amyloidosis [53].

P-tau181 has been incorporated as the (T) component of the AT(N) classification system, and plasma assays have reported an acceptable diagnostic performance [1,28,31]. However, its utility may be influenced by the clinical stage and analytical limitations. Despite the mentioned advantages, the recommended approach is to combine it with Aβ42 or t-tau to enhance the diagnostic accuracy [53].

The biomarker p-tau217 has demonstrated higher sensitivity and specificity in differentiating Alzheimer’s disease from other neurodegenerative disorders, particularly in the preclinical stages, compared to other plasma markers such as p-tau181 or Aβ42. As reported by Pais et al., p-tau217 showed AUC values above 0.93 in multiple cohorts using platforms such as Simoa^®^ (Quanterix Corp., Billerica, MA, USA) and Lumipulse^®^, outperforming t-tau, glial fibrillary acidic protein (GFAP), or neurofilament light chain (NfL), especially in studies with confirmation by neuroimaging and CSF biomarkers [53,54]. This isoform not only outperforms p-tau181 in early detection but also offers better differentiation from other tauopathies. As a result, p-tau217 is gaining momentum as the most promising biomarker for monitoring AD.

In addition to p-tau217, other emerging plasma-based biomarkers such as NfL and GFAP may offer supplementary diagnostic value [55,56]. The NfL biomarker indicates widespread axonal damage, thereby suggesting potential for monitoring neurodegeneration across various types of dementia [55]. Conversely, GFAP serves as a biomarker of astroglial activation, which may be elevated during the early stages of Alzheimer’s disease [56]. However, these biomarkers exhibit lower specificity for Alzheimer’s disease in comparison to tau-based markers; their integration into multi-biomarker panels may enhance the diagnostic precision [57]. This consideration is particularly relevant in Latin America, where the genetic, ethnic, and environmental heterogeneity present challenges to achieving diagnostic accuracy when relying solely on single biomarkers [57,58]. The relative diagnostic performance of plasma and CSF biomarkers is summarized in Table 1.

### 4.3. Progression of P-tau217 Across Braak Stages

The Braak staging system describes the topographic progression of the neurofibrillary tangles in AD. The classification includes six stages: from the transentorhinal cortex (stages I–II), progressing through the limbic regions (III–IV), and ultimately, reaching the associative neocortex [63].

Moloney et al. performed a digital immunohistochemical analysis post-mortem of the hippocampus in patients with AD. It was observed that the load of p-tau217 increased progressively along the stages, reaching statistically significant differences (*p* < 0.001): I (0.78%), II (0.70%), III (4.7%), IV (16%), V (13%) and VI (20%) [64]. In the subiculum, a vulnerable region, the p-tau217 load reached 38% in VI [64]. The progressive increase in p-tau217 across the Braak stages supports its value as a biomarker for monitoring the progression of AD [64]. In alignment with the pathological findings, an analysis based on the Braak stages, defined by tau-PET, observed that the levels of p-tau217 in the CSF increase significantly in stage II and reach a peak in p-tau217 in stage IV [65]. This isoform showed the most significant increases in the early stages compared with other isoforms of tau.

Feizpour et al. reported a progressive increase in p-tau217 in plasma across the Braak stages defined by tau-PET, with statistically significant differences from stage III onwards. Together, the anatomic progression of plasmatic p-tau217 and its association with structural changes influence their capacity to differentiate neocortical advanced stages [66]. These results support its accuracy as a peripheral biomarker sensitive to the topographic progression of the tau pathology, according to the Braak staging classification [66].

These advances emphasize the need to assess its applicability in different populational contexts, as well as to perform validity assessment and integration in diagnosis protocols and monitoring, especially in Latin American populations, where there is a lack of scientifically sound, contextualized evidence.

## 5. Comorbidities in Latin American Populations and Their Impact on Biomarkers

### 5.1. Contextual Barriers and the Need for Local Validation of AD Biomarkers in Latin America

Biomarker research has grown exponentially in Alzheimer’s disease over the last few years; however, most studies come from North American and European cohorts, which has generated a significant gap in the applicability to Latin American populations [3]. The region faces unique contextual challenges related to genetic diversity, inequalities, socioeconomic constraints, a lack of infrastructure, and the need to adapt clinical criteria and diagnostic tools, such as p-tau217, to local contexts [67]. Likewise, researchers have emphasized the importance of promoting studies in the Global South, ensuring that emerging biomarkers are validated for specific populations and health systems [68].

Health systems in Latin America face specific challenges influenced by the social determinants of health [69]. Inequity, poverty, lack of public policy, and limited access to healthcare attention and research investment create differential risks, hindering access to healthcare and the ability to apply new technology and diagnostic tools, such as plasma biomarkers in clinical settings [69]. While CSF and neuroimaging biomarkers have been demonstrated to be accurate and cost-effective in detecting changes at an opportune moment to treat AD, their routine use remains limited by high costs and lack of availability [3]. In this context, it is essential to validate p-tau217 in Latin American cohorts to ensure diagnostic accuracy and integrate it into the clinical and epidemiological reality of the region [3,67,68].

### 5.2. Regional Research: Studies in Latin America

This section outlines studies in Latin America that evaluate the diagnostic performance, methodological features, and limitations of p-tau217.

In Peru, a cross-sectional case-control study by Pandey et al., known as the GAPP (Genetic and Environmental Risk Factors for AD in Peru) study (N = 525; 234 healthy controls), utilized the Quanterix HD-X Simoa^®^ ALZpath v2 platform. The AUC without adjustment was 0.685 (95% CI: 0.62–0.75), with a sensitivity of 69.1% and a specificity of 71.8% [70]. After adjustment for sex, age, education, and *APOE* ε4 allele status, the AUC improved to 0.828 (95% CI: 0.77–0.87), with a sensitivity of 83.2% and a specificity of 70.5% [70]. Despite the promising findings, the study lacked confirmation by PET or CSF biomarkers, and the variability in the recruitment sites (urban vs. rural context) may have influenced the detection of preclinical cases [70].

In Colombia, the API consortium (Antioquia Registry of Patients with Early Onset AD) evaluated carriers of the PSEN1 E280A mutation [10]. This cross-sectional study by Palmqvist et al. (N = 622; 257 were non-carriers) found significant increases in the plasma p-tau217 levels by age 24.9 among carriers. A negative correlation was observed with the Mini-Mental State Examination test (ρ = −0.28, *p* = 0.02) and with the CERAD (Consortium to Establish a Registry for AD) cognitive score (ρ = −0.34, *p* = 0.003) in symptomatic carriers [10]. Diagnostic performance metrics (AUC, sensitivity, or specificity) were not reported. The method was a prototype immunoassay developed by Lilly Research Laboratories (Indianapolis, IN, USA), based on the Simoa^®^ platform, and no clinical cutoff point was defined [10].

Another study in Colombia, conducted by Aguillón et al. in the COLBOS cohort (N = 44), also evaluated carriers and non-carriers of PSEN1-E280A with a mean follow-up of 7.6 years. The baseline p-tau217 levels correlated positively with the future Aβ-PET and tau-PET uptake in multiple cortical regions and negatively with the Mini-Mental State Examination test scores (ρ = −0.57) [71]. However, formal diagnostic metrics were not reported, and the limitation to a single genetic mutation restricts the generalizability of the findings [71].

In Brazil, a study by Santos et al. at the IDOR Memory Clinic (N = 145) assessed a longitudinal cohort (follow-up duration 4.7 years) that included cognitively unimpaired individuals (CU) and those with amnestic mild cognitive impairment, AD, dementia with Lewy bodies, and vascular dementia (VaD). Using the Simoa^®^ HD-X platform, the same study reported an AUC of 0.94 (95% CI: 0.88–1.00) for distinguishing CSF biomarker positivity, although the sensitivity and specificity were not reported [72]. Higher p-tau217 levels were associated with an increased risk of diagnostic conversion (*p* = 0.0337) [72]. The limitations included the lack of PET confirmation, partial CSF sample availability, and cohort heterogeneity (differences in age, comorbidities, and a modest sample size) [72].

Another study in Brazil, conducted by Borelli et al. at the Hospital de Clínicas de Porto Alegre (N = 59), compared CU individuals, AD patients, and VaD patients. They reported significantly higher median p-tau217 levels in the AD group (0.77 pg/mL) than in the controls or VaD patients [73]. The AUC was 0.96 (95% CI: 0.91–1.00) for CU vs. AD, and 0.90 (95% CI: 0.78–1.00) for CU vs. VaD [73]. Correlations were also found with the CSF Aβ1–42/Aβ1–40 ratio (ρ = −0.49, *p* < 0.001) and t-tau levels (ρ = 0.38, *p* < 0.001) [73]. The limitations included a small sample size, the absence of PET imaging, and a lack of *APOE* genotyping [73].

Although only a small number of studies have evaluated p-tau217 in Latin America [10,70,71,72,73], most studies share significant methodological limitations that diminish their translational value. Several cohorts are restricted to urban populations or genetically similar groups, such as carriers of the PSEN1-E280A mutation [10,71], which can introduce selection bias and limit the generalizability to more diverse populations. Small sample sizes [71,73] and the absence of standardized cutoff values [10,72,73] weaken the statistical power and reproducibility. Only a few studies include longitudinal follow-up [71,72], restricting the understanding of the biomarker’s role in tracking disease progression. Notably, none of the studies to date have evaluated the cost-effectiveness or feasibility of implementing p-tau217 testing in clinical settings across low- and middle-income regions [10,70,71,72,73].

### 5.3. Influence of Comorbidities on Plasma P-tau217 Levels

While p-tau217 shows promise as a diagnostic biomarker, its interpretation may be affected by common comorbidities in Latin America. Some studies have examined the influence of hypertension, diabetes, chronic kidney disease, and obesity [74]. These conditions can potentially alter the biomarker levels or their clinical significance [74]. Pichet et al. found no significant differences in the p-tau217 levels between individuals with or without hypertension or diabetes [75].

Conversely, Mielke et al. reported elevated levels of p-tau217 in patients with chronic kidney disease, although the increase did not reach statistical significance. These findings suggest that while comorbidities may have a modest impact, more research is needed to determine their effect on biomarker reliability [76]. Figure 2 summarizes the interaction between comorbidities and the impact on diagnostic accuracy.

Comorbidities such as arterial hypertension and chronic kidney disease (CKD) have been associated with elevated plasma levels of p-tau217. In particular, the presence of CKD could influence the concentration of this biomarker due to a decrease in protein clearance, which would alter its clinical interpretation in patients with renal dysfunction [75].

On the other hand, Olvera et al. examined the modifying effect of type 2 diabetes on Alzheimer’s plasma biomarkers, finding that this condition may limit the ability of these biomarkers to reflect the brain amyloid and tau burden accurately. Although its impact on the overall diagnostic accuracy is minor, its high prevalence in Latin America makes it a clinically relevant factor to consider in future regional validation studies [77]. Additionally, socio-cognitive variables such as the educational level should be considered, as individuals with higher levels of education tend to possess a greater cognitive reserve, which may delay the clinical manifestation of the disease despite a significant neuropathological burden [70]. This dissociation could lead to underestimates of the impairment in individuals with a low educational level when the p-tau217 results are interpreted without appropriate adjustments. Furthermore, individuals with major depressive disorders may exhibit elevated p-tau levels even in the absence of cognitive impairment, suggesting a possible interference in the diagnostic specificity of the biomarker [78].

Although no direct correlations between inflammatory markers and p-tau217 have been conclusively established, a recent proteomic study by Zeng et al. reported that p-tau217 and several cytokines and chemokines (e.g., IL-5, CCL2, IL-4) showed parallel associations with longitudinal increases in the amyloid PET burden. These findings suggest that systemic inflammation may contribute to early Alzheimer’s pathophysiology and could indirectly influence the interpretation of the p-tau217 levels in the preclinical stages [79].

Given the potential influence of these clinical and sociodemographic variables, future studies must incorporate analyses stratified by comorbidities and contextual factors. In addition, the inclusion of adjusted multivariate models is recommended to assess the true diagnostic value of p-tau217 in populations with high clinical heterogeneity, such as those in Latin America.

### 5.4. Challenges and Opportunities for Implementation

Despite the technological progress, the clinical implementation of p-tau217 in Latin America faces notable barriers [68]. This biomarker is not yet commercially available in the region, and existing studies are limited by the small sample sizes and lack of representativeness [68]. Larger, multi-site cohorts across diverse populations are needed to validate its diagnostic performance and define reliable clinical thresholds [80]. In addition, when applied and reproducible, clinicians must be aware of factors that may introduce bias in interpretation, including renal function, age, body mass index, and the *APOE* ε4 genotype [80,81].

In addition to regional challenges, it is critical to recognize the technical limitations of p-tau217 assays. Analytical variability poses a concern due to differences between laboratories, reagent lots, analytical platforms, and individual biological fluctuations [57]. The preanalytical conditions, including sample handling, centrifugation procedures, and freeze-thaw cycles, can impact the measurement accuracy [57]. Manual platforms tend to show greater variability, while automated platforms typically have a lower coefficient of variation [57]. As a result, this variability complicates the standardization of universal cutoff values [82]. Currently, there are no universally accepted thresholds for p-tau217 in plasma because the values fluctuate significantly depending on the cutoff, analytical methodology, and measurement units [82]. Although some commercial assays report low cross-reactivity with other phosphoepitopes, independent validation is necessary [83]. Finally, although the cost of plasma p-tau217 (approximately USD 194 per assay on the Simoa^®^ platform) is lower than that of PET imaging, its use may be restricted in resource-limited settings [84]. It is essential to consider the institutional fees and infrastructure limitations in the region.

Although the recent 510(k) clearance by the FDA of the Lumipulse^®^ G pTau217/Aβ42 plasma assay [49] represents a relevant advance in the early diagnosis of AD, its clinical implementation in Latin America remains uncertain and probably delayed. This approval does not guarantee its availability in the region, where obstacles such as the regulatory fragmentation, budgetary constraints, and uneven diagnostic infrastructure persist. Each country must conduct its evaluation process through agencies such as COFEPRIS (Mexico City, Mexico), ANVISA (Brasília, DF, Brazil), or INVIMA (Bogotá, Colombia). Moreover, public health coverage rarely includes advanced biomarkers, which limits equitable access. Therefore, FDA approval should be seen as a starting point, not as an indicator of immediate implementation in resource-limited settings.

In support of its clinical potential, Lehmann et al. conducted a multicenter cohort study over three years. They found that elevated plasma p-tau217 levels were associated with a higher risk of dementia, reinforcing its potential utility in early detection among cognitively impaired individuals [80]. Overcoming these challenges will help reduce the diagnostic disparities and expand access to non-invasive, cost-effective tools for Alzheimer’s disease diagnosis in underrepresented regions [85].

## 6. Conclusions

Plasma p-tau217 has emerged as a promising diagnostic biomarker for the early detection of AD, demonstrating high sensitivity and specificity for early neuropathological changes. Nevertheless, most current evidence comes from European and North American cohorts, which limits its generalizability and applicability in other regions. Therefore, it is important to promote research in Latin American populations. In addition, regional factors, including the high prevalence of comorbid conditions, educational attainment, healthcare access, and socioeconomic status, must be considered to ensure that this biomarker is used in a precise, equitable, and culturally relevant manner. Advancing in all these aspects will be essential to ensure that p-tau217 provides evidence-supported benefits to individuals living with dementia or at risk of developing it in Latin America.

## Figures and Tables

**Figure 1 ijms-26-06633-f001:**
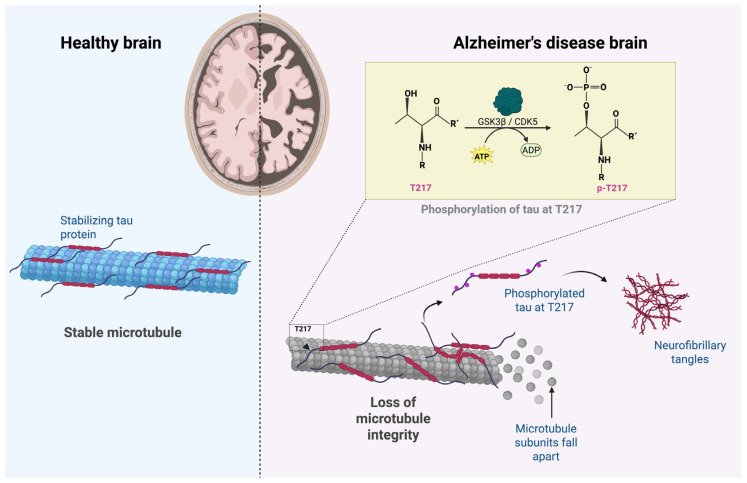
Disruption of microtubule integrity by tau phosphorylation in Alzheimer’s disease. Tau phosphorylation at T217 promotes microtubule destabilization and neurofibrillary tangle formation in Alzheimer’s disease. Created with BioRender.com.

**Figure 2 ijms-26-06633-f002:**
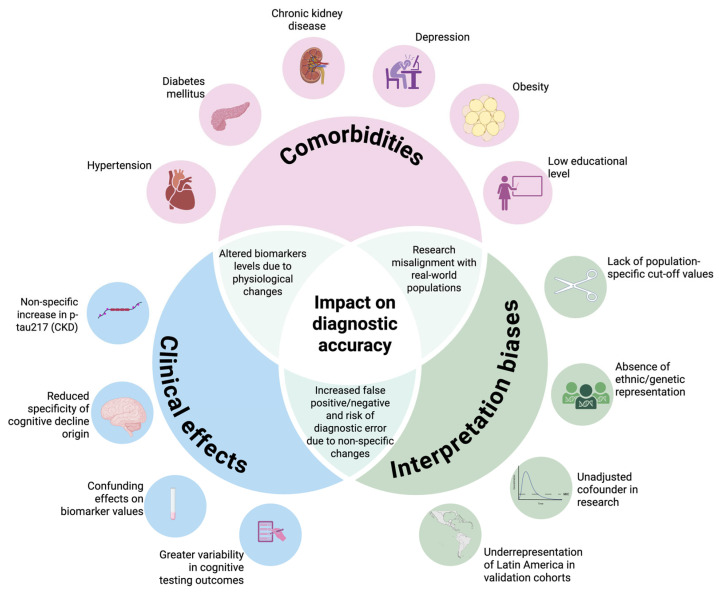
Challenges in terms of p-tau217 validation in Latin America. This Venn diagram illustrates how common comorbidities—such as chronic kidney disease, diabetes, and depression—and a low educational level may elevate the plasma p-tau217 levels or mask clinical symptoms through cognitive reserve. These overlapping influences can lead to misinterpretation (false positives or negatives), especially in underrepresented Latin American populations, highlighting the need for locally validated cutoffs and stratified analyses. Created with BioRender.com.

**Table 1 ijms-26-06633-t001:** Comparative diagnostic performance of p-tau217 and conventional Alzheimer’s disease biomarkers in CSF and plasma.

Biomarker	Platform/Assay	Matrix (CSF/Plasma)	AUC (95% CI)	Sensitivity (%)	Specificity (%)	Limitations	Reference
p-tau217	Simoa^®^ ALZpath p-tau217	Plasma	0.94 (0.89–0.99)	92	85	Variability between analytical platforms	[59]
p-tau181	Simoa^®^ p-tau181 Advantage V2	Plasma	0.889 (0.851–0.928)	86.6	80	Lower performance than p-tau217	[60]
t-tau	Simoa^®^ Neurology 3-Plex A	Plasma	0.505 (0.387–0.622)	91.8	23.1	High sensitivity, but very low specificity	[61]
Aβ42	Simoa^®^ Aβ42 Advantage V2	Plasma	0.539 (0.429–0.650)	56.5	38.5	Weak correlation, nonspecific and variable	[61]
p-tau217	MSD-based immunoassay (Lund University)	CSF	0.95 (0.88–1)	100 *	87.5 *	Low assay robustness and possible cross-reactivity with other tau isoforms	[62]
p-tau181	Lumipulse^®^ G p-tau181	CSF	0.954 (0.931–0.978)	91.8	90.5	Invasive and requires Aβ42 quotient for greater accuracy	[60]
t-tau	Lumipulse^®^ G total tau	CSF	0.870 (0.801–0.939)	76.1	89.7	Nonspecific marker	[61]
Aβ42	Lumipulse^®^ G β-Amyloid 1-42	CSF	0.911 (0.856–0.965)	90.9	79.5	High preanalytical variability	[61]

* The sensitivity and specificity values were derived from reported false positive and false negative rates in tau-PET-confirmed cohorts where not explicitly stated. The AUC, cutoff points, and limitations are included when available from the original studies. Data reflects the results from heterogeneous cohorts and assay platforms; interpretation should consider the methodological differences. Abbreviations: AUC: area under the curve; CI: confidence interval; CSF: cerebrospinal fluid; p-tau: phosphorylated tau; t-tau: total tau; Aβ42: amyloid beta 1-42; MSD: Meso Scale Discovery; Simoa^®^: Single Molecule Array; 3-Plex: three-analyte multiplex assay.

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
