# Peer review of "P-tau217 as a Biomarker in Alzheimer’s Disease: Applications in Latin American Populations"

_ijms, 2025, doi:10.3390/ijms26146633_

Round 1
Reviewer 1 Report
Comments and Suggestions for Authors
Remarks about the manuscript:
The authors have presented a review article, "P-tau217 as a Biomarker in Alzheimer’s disease:
Applications in Latin American Populations." The article focuses on the development of a new marker, P-tau217, for the early detection of Alzheimer’s disease in Latin America, where conventional resources are not universally available. This manuscript presents a carefully selected set of reference articles. It details current experimental outcomes and is supported by clear explanations. Relevant references comparing well-known markers such as p-tau181, t-tau, and Aβ42 have been included, despite the limited information available about them in the Latin American population.
Overall, the manuscript is well-written in English, with clear and concise sentences. The present manuscript contains most of the data required for publication in the IJMS journal; therefore, I will recommend it for publication in its current form.
This manuscript (ID: IJMS-3717851) can be accepted without revision.
Author Response
We sincerely thank Reviewer 1 for their positive assessment and thoughtful comments. We appreciate the recognition of the manuscript's strengths and are grateful for the support toward its publication.
Reviewer 2 Report
Comments and Suggestions for Authors
The manuscript provides a comprehensive narrative review of p-tau217 as a biomarker for Alzheimer’s disease, with a notable focus on Latin American populations. It is well-organised and cites a broad range of contemporary literature. The inclusion of region-specific data is valuable and timely. However, several aspects could be strengthened to improve scientific rigour, balance, and clarity.
Comments and suggestions
- The narrative repeatedly stresses p-tau217’s superiority over other biomarkers (e.g., p-tau181, Aβ42) but does not sufficiently critique its limitations, such as assay variability, lack of standardised cut-offs, and cross-reactivity risks. The review would benefit from a more balanced discussion, explicitly addressing variability across analytical platforms, Limited accessibility of advanced assays in low-resource settings, and potential confounding due to comorbidities beyond kidney disease (e.g., hepatic dysfunction, inflammatory states).
- While regional studies are summarised, critical appraisal of their methodological limitations is somewhat superficial. Encourage the authors to discuss potential selection biases (e.g., urban vs. rural, single-mutation focus), reflect on the impact of small sample sizes and the lack of longitudinal validation in most studies and highlight the absence of cost-effectiveness studies for p-tau217 deployment in Latin America.
- The manuscript appears to be a narrative review, but does not describe the search strategy, Inclusion/exclusion criteria, time frame for the literature considered, or risk of bias assessment. This weakens the reproducibility and transparency of the synthesis. A brief methodological section would enhance scientific credibility.
- The discussion of comorbidities (e.g., hypertension, diabetes) is relatively superficial. Could you include a more nuanced evaluation of confounders through cited meta-analyses or regional epidemiological data, and emphasise the need for stratified analyses in future studies?
- The authors focus on p-tau217 but do not engage sufficiently with other emerging plasma biomarkers (e.g., neurofilament light chain, GFAP) or multi-biomarker approaches that may offer complementary value in Latin America.
- Figures and Tables: Venn diagrams are visually apparent, but the legends could provide more interpretive depth rather than descriptive captions.
- Table 1 does not consistently report confidence intervals, sensitivity/specificity, or assay limitations across entries. A more standardised format would aid clarity.
- Several statements could benefit from clearer qualification, e.g., “p-tau217 has a superior sensitivity and specificity…” in what settings, under what conditions?
- The manuscript refers to FDA clearance of assays; the implications for Latin America are unclear and should be elaborated.
- Citations could be integrated more tightly to support specific claims (e.g., statements regarding the impact of comorbidity).
Author Response
The manuscript provides a comprehensive narrative review of p-tau217 as a biomarker for Alzheimer’s disease, with a notable focus on Latin American populations. It is well-organised and cites a broad range of contemporary literature. The inclusion of region-specific data is valuable and timely. However, several aspects could be strengthened to improve scientific rigour, balance, and clarity.
Comments and suggestions
- The narrative repeatedly stresses p-tau217’s superiority over other biomarkers (e.g., p-tau181, Aβ42) but does not sufficiently critique its limitations, such as assay variability, lack of standardised cut-offs, and cross-reactivity risks. The review would benefit from a more balanced discussion, explicitly addressing variability across analytical platforms, Limited accessibility of advanced assays in low-resource settings, and potential confounding due to comorbidities beyond kidney disease (e.g., hepatic dysfunction, inflammatory states).
Answer: We thank the reviewer for this important observation. In response, we expanded section 5.4 “Challenges and Opportunities for Implementation” to include a detailed discussion of the technical limitations of p-tau217. This includes inter-laboratory variability, preanalytical handling, lack of standardized cutoffs, cross-reactivity concerns, and cost-related barriers to implementation in low-resource settings. These additions aim to provide a more balanced view of the biomarker's current limitations and support the need for cautious regional adaptation. Relevant citations were incorporated to support these points (Line 486-499).
About the potential confounding comorbidities, we performed an updated literature search focused on hepatic and inflammatory comorbidities. While no studies were found regarding hepatic dysfunction and p-tau217, a recent proteomic analysis (Zeng et al., 2024) identified parallel associations between plasma p-tau217 and multiple inflammatory markers (e.g., IL-5, IL-4, CCL2) with longitudinal increases in Aβ PET burden. Although direct correlations were not reported, this evidence supports the hypothesis that peripheral inflammation could indirectly modulate p-tau217 levels and warrants further investigation in heterogeneous populations. We have now incorporated this into section 5.3 of the manuscript (Line 466-471).
- While regional studies are summarised, critical appraisal of their methodological limitations is somewhat superficial. Encourage the authors to discuss potential selection biases (e.g., urban vs. rural, single-mutation focus), reflect on the impact of small sample sizes and the lack of longitudinal validation in most studies and highlight the absence of cost-effectiveness studies for p-tau217 deployment in Latin America.
Answer: Thank you for this valuable suggestion. We have incorporated a critical paragraph at the end of Section 5.2 ("Regional research: Studies in Latin America") discussing key methodological limitations of the studies conducted in Latin America. Specifically, we address selection biases related to genetically homogeneous cohorts, small sample sizes, lack of standardized cutoffs, limited longitudinal data, and the absence of cost-effectiveness evaluations. These additions aim to improve the balance and critical depth of our regional analysis. (Line 400-409).
- The manuscript appears to be a narrative review, but does not describe the search strategy, Inclusion/exclusion criteria, time frame for the literature considered, or risk of bias assessment. This weakens the reproducibility and transparency of the synthesis. A brief methodological section would enhance scientific credibility.
Answer: Thank you for the observation. We have now added a brief methodological section clarifying that the review followed SANRA recommendations. The search was conducted in PubMed from 2020 to mid-2025 using MeSH terms, DeCS descriptors, and free-text keywords related to Alzheimer’s disease, p-tau217, biomarkers, comorbidities, and Latin American populations. Inclusion criteria focused on original and review articles with clinical relevance; animal studies and articles without biomarker data were excluded (Line 65-80).
- The discussion of comorbidities (e.g., hypertension, diabetes) is relatively superficial. Could you include a more nuanced evaluation of confounders through cited meta-analyses or regional epidemiological data, and emphasise the need for stratified analyses in future studies?
Answer: Thank you for this valuable suggestion. We have expanded the discussion to provide a more comprehensive evaluation of the potential influence of comorbidities such as arterial hypertension, chronic kidney disease, type 2 diabetes, and major depressive disorder on plasma p-tau217 levels. Additionally, we now address the potential impact of cognitive reserve, particularly educational attainment, on biomarker interpretation in Latin American populations. This section highlights the relevance of stratified analyses and multivariate adjustment in future research to improve the diagnostic utility of p-tau217 in diverse clinical contexts. These additions can be found in section 5.3 (Line 448-465 y 472-476).
- The authors focus on p-tau217 but do not engage sufficiently with other emerging plasma biomarkers (e.g., neurofilament light chain, GFAP) or multi-biomarker approaches that may offer complementary value in Latin America.
Answer: Thank you for this valuable comment. We agree that emerging biomarkers such as neurofilament light chain (NfL) and glial fibrillary acidic protein (GFAP) are relevant in the broader context of neurodegenerative diseases. However, these markers are not specific to Alzheimer’s disease, as they are elevated in several other neurological conditions. Since the aim of our review is to highlight the specificity and clinical potential of p-tau217 for early detection and monitoring of Alzheimer’s disease, we opted to maintain a focused discussion. Nevertheless, we have now added a brief paragraph in the discussion section acknowledging the complementary role of NfL and GFAP and noting that p-tau217 remains the most disease-specific plasma biomarker for AD to date (Line 287-296).
- Figures and Tables: Venn diagrams are visually apparent, but the legends could provide more interpretive depth rather than descriptive captions.
Answer: Thank you for the suggestion. We revised the legend of the Venn diagram to more accurately reflect the content discussed in the section "Influence of comorbidities on plasma p-tau217 levels." The updated legend now emphasizes how specific comorbidities can increase biomarker levels or interfere with clinical interpretation—particularly in populations with low cognitive reserve or renal dysfunction—and how these effects may bias diagnosis in underrepresented Latin American cohorts (Line 442-447).
- Table 1 does not consistently report confidence intervals, sensitivity/specificity, or assay limitations across entries. A more standardised format would aid clarity.
Answer: Thank you for this observation. We have revised Table 1 to include a standardized format with clearly defined columns: Biomarker, Platform / Assay, Matrix (CSF / Plasma), AUC (95% CI), Sensitivity (%), Specificity (%), Limitations, and Reference. For studies where sensitivity or specificity were not directly reported, we derived the values from available false positive/negative rates when appropriate and noted this in the table legend. We believe these changes enhance the table's clarity and comparability (Line 298).
- Several statements could benefit from clearer qualification, e.g., “p-tau217 has a superior sensitivity and specificity…” in what settings, under what conditions?
Answer: Thank you for this valuable observation. We revised the relevant paragraph to clarify the specific conditions under which p-tau217 demonstrates superior diagnostic performance, including comparisons across cohorts, disease stages, and biomarker platforms. These changes are now reflected in Section 4.2, prior to Table 1 (Line 278-284).
- The manuscript refers to FDA clearance of assays; the implications for Latin America are unclear and should be elaborated.
Answer: We agree that FDA approval does not necessarily imply clinical availability in Latin America. To address this, we added a paragraph outlining the region-specific regulatory, economic, and logistical challenges that may delay or limit implementation. This addition can be found as the penultimate paragraph of Section 5.4 (Line 500-509).
- Citations could be integrated more tightly to support specific claims (e.g., statements regarding the impact of comorbidity).
Answer: We carefully reviewed the manuscript and improved citation alignment with statements, especially in sections 4.1 and 4.3, ensuring each claim is now directly supported by a cited reference (Line 347 and 350, 413 and 420).